# Job Satisfaction in Remote Work: The Role of Positive Spillover from Work to Family and Work–Life Balance

**DOI:** 10.3390/bs13110916

**Published:** 2023-11-10

**Authors:** Elizabeth Emperatriz García-Salirrosas, Rafael Fernando Rondon-Eusebio, Luis Alberto Geraldo-Campos, Ángel Acevedo-Duque

**Affiliations:** 1Faculty of Management Science, Universidad Autónoma del Perú, Lima 15842, Peru; 2Department of Humanities, Universidad Privada del Norte, Lima 15314, Peru; rafael.rondon@upn.edu.pe; 3Dirección de Investigación e Innovación, Escuela Superior La Pontificia, Ayacucho 05002, Peru; luisgeraldo@elp.edu.pe; 4Programa de Doctorado en Ciencias Sociales, Universidad Autonóma de Chile, Santiago 7500912, Chile; angel.acevedo@uautonoma.cl

**Keywords:** family-supportive supervisory behaviors, work-to-family positive spillover, work–life balance, job satisfaction in remote work

## Abstract

The objective of this research is to propose and validate a theoretical model that explains job satisfaction in remote work influenced by family-supportive supervisory behaviors (FSSBs) and, in addition, to evaluate the mediating role of work-to-family positive spillover (WFPS) and work–life balance (WLB) in this influence. A non-experimental cross-sectional study was conducted using a self-administered survey to a sample of 396 teleworkers in Lima, Peru. The hypothesized model was analyzed using PLS-SEM based structural equation modeling. The results show that FSSB has a direct effect on both job satisfaction in remote work and WFPS and WLB. In addition, it shows that WFPS and WLB have positive effects on job satisfaction in remote work. Also, the results show that WFPS and WLB have a mediating role in the influence of FSSB on job satisfaction in remote work. In conclusion, this study highlights the importance of supervisor behavior, positive work-to-family spillover, and work–life balance in remote workers’ job satisfaction. It is suggested that companies adopt policies and practices that encourage work–life balance as well as a favorable supervisory environment.

## 1. Introduction

The phenomenon of remote work has shown a constant upward trajectory in the last decade, driven, above all, by technological advances and the growing need for flexibility in the workplace. The COVID-19 outbreak caused companies that had not yet implemented remote work to quickly adjust their processes to maintain operations. The aforementioned modifications caused an increase in emotional exhaustion and work exhaustion, which affected the quality of life, mental well-being, and job satisfaction of employees [1]. The aforementioned issues arose as a consequence of the transition to remote work, as employees faced challenges related to family obligations and external circumstances that influenced their levels of job satisfaction and performance [2]. However, the possibility of working remotely has facilitated a greater level of flexibility in managing professional and family obligations. Consequently, people who have successfully adapted to this type of work feel more satisfied with their jobs.

Job satisfaction is a crucial element of the professional sphere, as it has the potential to influence the productivity, dedication, and retention of employees within an organization [3]. Therefore, it is vital to understand the impact of remote work on employee job satisfaction. Scientific research has examined the elements that influence employee job satisfaction in the context of remote work. These characteristics include, but are not limited to, family-supportive supervisory practices, positive work-to-family spillover, and work–life balance [4].

A positive spillover from work to family encompasses the emergence of positive experiences and emotions in the professional environment, which then exert a favorable influence on the individual’s family environment [5]. This issue has aroused the interest of scholars, as both entities and individuals recognize the inherent link between the professional and family spheres. The transfer of beneficial components from work to the family environment is facilitated by numerous mediating mechanisms [6]. Several elements, such as flexible work schedules and activities, supportive work environments, and the acquisition of coping mechanisms for work-related stress, contribute to the establishment of a positive cycle in which the work environment enhances family life. In this context, addressing mediating factors within companies helps to facilitate a more favorable work–life balance for employees [7].

The concept of positive spillover from work to family is a topic of significant interest in various governmental, corporate, and social institutions, including those within the realm of higher education. This is because understanding the effects of such spillover on the interplay between the work and home domains is crucial for the overall well-being of individuals [8]. This concept recognizes how positive emotions, healthy work relationships, and job satisfaction can influence family dynamics. Mediating factors, such as job autonomy, social support, and the ability to manage stress, act as bridges that allow these positive experiences to transcend from one environment to the next [9]. In a globalized and connected world, understanding and promoting the positive spillover from work to family becomes fundamental for the comprehensive well-being of people and the construction of more balanced societies.

The above is representative when the balance between work and personal life is studied at the same time; these variables are essential to guarantee satisfaction in remote work [10]. As more people adopt this type of work, there is a need to establish clear boundaries between work responsibilities and time dedicated to personal life. Achieving the right balance contributes significantly to employee productivity, well-being, and satisfaction [11]. To encourage this balance, it is essential to establish defined schedules for work and free time. Establishing clear boundaries about when the workday starts and ends helps to prevent burnout and maintain separation between the two [12]. Additionally, communicating effectively with colleagues and supervisors about available times and downtime helps to set realistic expectations and avoid unnecessary interruptions outside of work hours [13].

In Latin America, the phenomenon of positive work-to-family spillover and work–life balance takes on particular relevance due to the cultural and socioeconomic characteristics of the region [14]. The importance of family relationships and the need to maintain a balance between work and family commitments are intrinsic aspects of daily life. Mediating factors are especially important in this context, since they can facilitate a smooth transition from work emotions and satisfaction to the family environment. The promotion of work practices that promote well-being and harmony between both areas can have a profound impact on the quality of life of people in Latin America [15].

Despite the significance of this research area, there has been a limited number of studies conducted in the Latin American region. The existing research indicates that there exists a multifaceted connection between work and family contacts, which can be identified as a determining factor in the overall well-being of employees. By fostering a favorable environment that acknowledges these intermediary factors, employees in Latin America have the ability to attain a higher level of harmony between their work and family domains, resulting in an enhancement in their overall quality of life [16,17,18,19].

Based on the above, the objective of this research is to propose and validate a theoretical model that explains job satisfaction in remote work, influenced by family-supportive supervisory behaviors and, in addition, to evaluate the mediating role of work-to-family positive spillover and work–life balance in this influence. 

Regarding the organizational framework of this manuscript, it commences with Section 1, the introduction, wherein the problematic landscape is delineated and the hypotheses are substantiated through a comprehensive literature study. Subsequently, Section 2, titled “Materials and Methods,” provides a comprehensive account of the methodologies, techniques, and tools employed for the purpose of data analysis. Section 3 thereafter offers the contrasting outcomes that pertain to the stated hypotheses. Next, Section 4 of the paper presents a comprehensive analysis of the findings, including a discussion of their implications, limitations, and potential for future research. Lastly, the following section, denoted as Section 5, is presented with the objective of elucidating the conclusions derived from the present study.

### 1.1. Literature Review and Hypothesis

#### 1.1.1. Job Satisfaction in Remote Work

Remote work has taken on great relevance as a result of the COVID-19 health crisis, which accelerated its mandatory implementation in almost all organizations in the world, being the main complement to work activity for economic growth in the times of the pandemic [1,20]. As a result of the sudden change, this type of work was associated with other contextual variables and the consequences of the adaptation process of work at home, which caused alterations in workers’ emotional states and job satisfaction, impacting family relationships and the mental health of workers [21,22,23].

However, the remote work modality is not recent; its origins date back to the 1970s, when engineer Jack Nilles coined the term to refer to a work modality where employees can carry out their activities outside the office, relying on telecommunications technologies. This idea was reinforced by Alvin Toffler who, in 1980, realized that the technological revolution would cause a change in the conception of the workplace, which would move to places other than the physical space of the company [24,25,26]. Although its adoption was limited at the beginning, by the 1980s and 1990s, it began to expand, facilitated by advances in computing and the Internet [25,27]. However, its growth was relatively slow until the second decade of the 21st century, when digital connectivity improved significantly. Likewise, coronavirus marked a turning point in this type of employment, increasing the transition to remote work for activities that did not require physical presence, and even after the economic reactivation, this is still the new normal [1].

Conceptually, teleworking is defined as a flexible way of working that allows the performance of work activities to take place outside the company, without requiring the physical presence of the employee, through the use of information and communication technologies [24,27]. Its modalities include working from home, or work that requires movement between different places other than the office or home [23].

From a theoretical approach, various perspectives have been used to study teleworking and its implications. In this way, the theory of the work–family boundary states that both systems contain unequal norms and demands that influence each other. However, remote work blurs the spatial and temporal boundaries between the work and personal spheres, creating challenges for its members to maintain a healthy balance that is difficult to achieve [28]. Likewise, with regard to interactions between members of an organization, the media richness theory suggests that remote communication involves more ambiguity and coordination difficulties than face-to-face interaction, so communication problems are more likely to exist if there is no decoding of all aspects of the language that occurs in face-to-face interactions [29]; hence, the strategies for implementing remote work include actions that reduce these communication distances as much as possible. For its part, the theory of social presence indicates that the lack of social signals in virtual environments can reduce the cohesion of work groups, since presence in an interaction allows for more fluid feedback to clarify objectives and establish agreements [30]. Although these theoretical models explain the difficulties faced by a remote work model, they have also provided the guidelines for consideration to establish it more effectively in current times, since technological advances allow us to have the tools to develop activities with greater dynamism in communication and with greater control of work times outside the physical spaces of the company [1,27,31].

Regarding job satisfaction, this concept refers to the positive attitude and motivation that a person feels towards their job [32,33]. It is a complex concept that encompasses several factors such as salary, working conditions, relationships with colleagues and superiors, and professional development, among others [1,34,35]. Likewise, it is a topic that has received extensive attention in organizational research due to its influence on employee performance and productivity [36,37,38]. Furthermore, it has been proven that job satisfaction is the attitude resulting from internal factors (values, personality, needs, desires, experiences) and external factors (salary, workload, social support, professional development, physical space) of the individual, related to one’s work activity and working conditions [1,39].

Research on job satisfaction dates back to the 1930s with the work of Hoppock and Spiegler, who developed one of the first scales to measure it, presenting a group of factors that intervene in job dissatisfaction such as fatigue, monotony, working conditions, and supervision [40]. Subsequently, Herzberg proposed the two-factor theory, which establishes that there are extrinsic factors (work conditions, salary, company policies) that only prevent dissatisfaction, but do not generate satisfaction, as well as intrinsic factors (responsibility, recognition, personal fulfillment) that do promote job satisfaction [41]. Another important model that explains job satisfaction is that of Hackman and Oldham’s job characteristics, who highlight five dimensions that influence satisfaction, presented as the variety of skills, the identity of the task, the meaning of the task, autonomy, and feedback [42].

Regarding the relationship between remote work and satisfaction, studies have found that this modality can increase worker satisfaction towards their work activities due to the control of the time and environment that they have in their activities [2]. Also, it has been found that teleworking can reduce stress and increase satisfaction, as well as regulate the balance between work and personal life [3]. However, these relationships are not conclusive, since the causal relationship between teleworking and satisfaction is diverse in its results, with other studies finding that workers under this modality show dissatisfaction with their activities due, among other reasons, to the disconnection from work, social labor relations, and the perception of a greater workload [43,44].

Research on work–family balance, family support monitoring behavior, and work happiness has been conducted in a variety of forms, from review studies to primary research. For example, a review study on the topic provides a comprehensive conceptual model of work–family support, indicating that work–family support laws have a favorable impact on family happiness, employee retention, job satisfaction, and job commitment. Furthermore, empirical research also demonstrates how family-friendly policies and flexible time management affect workers’ well-being and, consequently, their productivity [45,46].

#### 1.1.2. Family-Supportive Supervisory Behaviors (FSSBs) and Job Satisfaction in Remote Work

The interest in studying FSSBs arose from the need of many employees to balance their work and family responsibilities [47,48]. Thus, the first studies focused on exploring, from the workers’ perspective, what type of support they perceived from their supervisors to be able to effectively carry out their family role [49].

Like all variables, organizational behavioral researchers sought to more concretely conceptualize and measure the behaviors specifically exhibited by supervisors to provide such family support. One of the main contributions was the development by Hammer at the end of the first decade of the 2000s, who, together with other researchers, proposed a multidimensional scale that evaluates supportive supervisory actions such as the adjustment of work schedules, the reassignment of tasks to facilitate responsibilities towards family members, providing information resources, or becoming directly involved in finding solutions to work–family conflicts [50]. Likewise, the authors defined it as the actions exhibited by supervisors that are perceived by employees as indicators of support regarding their family life and family roles [50,51]. This conceptualization contrasts with supervisors who ignore their subordinates’ family responsibilities or who view family work as an impediment to job performance [52].

However, the bases of the FSSBs are found in theoretical models that explain the motivations that drive the exchange of transactions between people, in a cost–benefit analysis of the interactions they carry out [53,54]. Thus, “Social Exchange Theory” maintains that individuals evaluate the fairness of reciprocity between what they contribute in terms of effort and dedication, and what they receive in the form of support from both their family and their workplace. That is, when supervisors offer substantial support to employees in their family responsibilities, and they perceive that this support compensates for the emotional and effort costs associated with fulfilling these responsibilities, they are likely to have a higher level of job satisfaction. This theoretical model also exposes the concept of reciprocity and examines the way in which individuals react to the behaviors exhibited by their counterparts, with the objective of providing an equitable response to the one received [55,56]. Thus, when supervisors demonstrate support for employees’ family obligations, it is likely that employees will experience a feeling of duty to reciprocate by perceiving appreciation and concern for them. This phenomenon can be attributed to the fact that work connection provides individuals with emotional and psychological benefits, thereby increasing their level of dedication and overall well-being in the workplace.

However, FSSB research indicates that companies have the potential to effectively manage work and family obligations by implementing family-supportive behaviors on the part of supervisors and fostering a work environment that promotes the enrichment of work and family environments [47,52,57]. Family support workplace benefits encompass several elements, such as emotional support, work flexibility, the recognition of family obligations, and encouraging employees to prioritize their family commitments. Therefore, establishing a culture of work–family enrichment involves promoting work–family balance, providing family-oriented policies and benefits, and actively encouraging employees to use these resources [49,58]. In this regard, a study revealed that there was a positive correlation between FSSBs and work–family enrichment with job satisfaction, organizational commitment, and performance appraisal by the supervisor. Furthermore, there was a negative correlation between FSSBs and work–family enrichment with intention to resign and work–family conflict [59].

Studies that have attempted to ensure the impact of family-supportive supervisory behaviors on job satisfaction have included it in a group of variables to find mediation or moderation relationships between various terms associated with the work environment [47,52,60,61]. In this regard, the research reviewed found that employees who perceived greater support from their supervisors for work–family management reported significantly greater job satisfaction. However, supervisor support in aspects such as schedule flexibility, autonomy, and management of communication with the family were positively associated with employee satisfaction [13,47,48,49].

Furthermore, one study has provided confirmation of the autonomous mediating effects as well as the sequential mediating effects of perceived internal identity and affective commitment. To achieve this goal, the researchers constructed a comprehensive model that incorporates supervisor family-supportive behavior, perceived internal identity, affective commitment, and employee proactive behavior. The results of this study validate the sequence of effects, according to which the supervisor’s family support behavior initially influences perceived internal identity, which subsequently affects affective commitment and, finally, influences the proactive behavior of employees [62]. Due to the above, we intend to test the following hypothesis:

**H1.** 
*FSSBs have a positive influence on job satisfaction in remote work.*


#### 1.1.3. FSSBs and Work-to-Family Positive Spillover (WFPS)

In the field of work and family dynamics, the concepts of FSSB and WFPS have emerged as a very significant aspect to understand behavior and motivations at work, which is why both concepts have been exposed as the influence of experiences and emotions, including pleasant experiences generated in the workplace and in family life, but with differences in the resulting worker’s behavior. While FSSB is the result of the actions of the company’s climate in the family, the WFPS is the search for balance and the demarcation of work activities in the family, and vice versa [18,49,63]. In this regard, the theoretical model of work–family spillover proposes that there is the possibility of indirect effects when individuals encounter positive emotions in their workplace (such as satisfaction, pleasure, and joy) given that these can spread to their family life, leading to an increase in positive effects and general well-being in this area [64].

Indeed, WFPS is a concept that refers to the transmission of favorable experiences, feelings, and attitudes from the professional environment to the family environment, thus potentially improving the caliber of family connections and general satisfaction in daily existence [65]. In this regard, this concept is associated with a set of variables that drive motivation and improve performance at work [5,66]. Likewise, studies have identified a strong correlation between this variable and the level of autonomy in the workplace; that is, greater control and mastery of work activities leads to possible favorable effects at home [12,67].

However, research has revealed that supervisors’ behavior in providing support to employees’ families has the potential to generate positive spillover effects from work to home through work engagement [66]. The aforementioned results are consistent with previous academic research on the relationship between general supervisor support and improved work–family dynamics. Therefore, this study aims to answer the following hypothesis:

**H2.** 
*FSSB has a positive influence on the WFPS in remote work.*


#### 1.1.4. FSSB and Work–Life Balance (WLB)

WLB refers to an individual’s ability to adequately manage their professional and personal obligations, ensuring that the demands of both areas are adequately met without compromising either [35,68]. Achieving an optimal work–life balance has been shown to have positive effects on various aspects of people’s lives, including job satisfaction, mental well-being, physical health, and overall quality of life [69,70,71].

Although WLB is a topic of varied research to measure the effects of the organizational environment in the different groups where the worker interacts, the theoretical bases that fully define it are still under construction [23]. However, there is agreement on its effects, as well as the factors that develop an imbalance in this balance. Regarding the effects, a lack of work–life balance can have consequences that affect a person’s health, both in the workplace and in their psychological health. Thus, effects such as work stress, exhaustion, depression, and psychosomatic disorders, among others, have been described [72,73]. Regarding the risk factors that cause an imbalance in the family–work boundary, studies have presented evidence related to a negative perception of the work environment produced by the poor management of leaders, supervision problems, workload, a lack of commitment in the work environment, mistrust, and dysfunctional family environments [71,74,75,76,77]. Furthermore, regarding the relationship between FSSB and WLB, studies indicate that supervisors who provide support to employees’ families play a significant role in meeting their demands related to achieving work–life balance, which translates to a positive impact on work performance, since it improves psychological well-being and reduces the intention to resign from their positions [47,57]. Given the above, it is intended to resolve the following hypothesis:

**H3.** 
*FSSB has a positive influence on WLB in remote work.*


#### 1.1.5. WFPS and Job Satisfaction in Remote Work

It is important to evaluate how the WFPS can influence job satisfaction during the remote work day due to the change in modality in workers’ contracts. It has been found that job demands, stigma awareness, supervisor incivility, and negative work–family spillover affect burnout levels, while job satisfaction and WFPS may be relevant resources to prevent burnout [78]. Furthermore, people with a negative work–family spillover were found to be less satisfied with their lives, while those with a positive work–family spillover were more satisfied. However, when considering other predictors of life satisfaction, such as baseline health, sex, income, and personality, these became significant and were mediated by work and family satisfaction [79].

On the other hand, family satisfaction and career satisfaction tend to be correlated when work constraints are modest or when family identity is prominent [80]. However, the WFPS is related to work-related well-being and general well-being; WFPSs do not directly affect any well-being indicator [81]. On the other hand, the level of facilitation of work and family life and a more constructive style of interpersonal communication varies between work shifts, so it may indicate that a rapid rotation system is better for work and family and their relationships [82]. However, work demands and time pressure can negatively affect mental health in home-based workers through negative work–family stress. Along those same lines, high job control and workplace support can also improve mental health through the mediation of WFPS [83]. However, the negative and positive indirect effects of work on the family are different experiences. In contrast, work and family factors that facilitated development (e.g., decision-making freedom and family support) had less negative and more positive side effects on work and family, since work and family barriers (pressure and disagreements) had more negative and less positive indirect effects on work–family relationships [84].

From another perspective, it is clear that support can benefit women and men differently, underscoring the importance of a supervisor and an organization that balance work and personal life. For example, men with a leader who supports work–life balance have fewer negative effects, fewer intentions to leave work, and higher job satisfaction. Furthermore, teleworking benefits men more than women [85]. On the other hand, mothers’ positive outlook after work is associated with youths’ reports of more positive mood, better sleep quality, and longer sleep duration. Likewise, mothers with positive work experiences report less negative effects and fewer physical health symptoms after work [51]. Based on this background, we seek to resolve the following hypothesis:

**H4.** 
*WFPS has a positive influence on job satisfaction in remote work.*


#### 1.1.6. WLB and Job Satisfaction in Remote Work

It has been shown that work–family balance affects organizational pride and job satisfaction, but not intentions to quit [86]. On the other hand, work–life balance and work–family conflict improve employee performance. However, job satisfaction moderates the relationships between work–life balance, work–family conflict, and employees’ perceived performance [87]; in that same line of ideas, job satisfaction partially affects the relationship between work–life balance and retention [88], since the balance between work and personal life has a positive impact on job satisfaction [89].

Recent evidence reveals that work–life balance is negatively related to work stress, unlike the relationship with job satisfaction and work commitment. Furthermore, the work environment moderates the effect of work–life balance on stress and job satisfaction [90]. Along these lines, a previously published study showed that job satisfaction is achieved in two ways: first, a clear role at work, little ambiguity in the role, balance between work and private life, and an impact on job satisfaction. That is, work–life balance moderates the impact of relationships on job satisfaction. Second, a low role overload improves job satisfaction [91]. However, there is a mediating effect of WLB in the relationship between job satisfaction and job benefit [92]. Thus, office and home workers tend to have similar high levels of WLB support and job satisfaction; therefore, office workers report higher levels of work–life balance support than other types of workers and home workers, including those based on clients [93].

On the other hand, remote work does not have clear effects on job satisfaction, but it does negatively affect the balance between professional and private life. If the imbalance is due to private interests, it does not contradict the characteristics of the work, because those who work from home are happier than those who want to [68] work at home, have greater job satisfaction, and a better balance between work and family life with a strict contract than with a non-binding contract. Furthermore, a low quality of work–life balance is evident, as job satisfaction is similar for teachers with low, average, or high work–life balance; those with a high quality of work–life balance who had high employee well-being also had high job satisfaction [94]. Also, a high level of WLB is associated with job and life satisfaction in individualistic rather than collectivistic cultures, since high levels of work–life balance were associated with work and life satisfaction, but negatively associated with anxiety in egalitarian cultures. Finally, strong support is necessary so that work–life balance benefits diverse employees and culture as a moderator of these relationships [95].

From another approach, it is evident that professional careers have a significant impact on organizational citizenship behavior (OCB) and WLB. Thus, WLB and job satisfaction may indicate a critical link between career and organizational citizenship behavior [96]. Another study indicates that workers claim to have difficulties reconciling work and family, and one-third consider leaving their job for these reasons [97]. It is also evident that organizational commitment is not related to professional–personal balance. In contrast, organizational commitment mediates the relationship between job satisfaction and the organizational environment, but not professional–personal balance [34]. Based on the above, we seek to answer the following hypothesis:

**H5.** 
*WLB has a positive influence on job satisfaction in remote work.*


#### 1.1.7. FSSB, WFPS, and Job Satisfaction in Remote Work

Numerous studies have looked into the relationship between FSSB, WFPS, and job satisfaction when it comes to telecommuting. For instance, it has been discovered that positive relationships between job autonomy and supervisor family support behaviors, as well as work–life balance and job satisfaction, are modified by beneficial work-to-family spillovers and prior telecommuting experience [2]. While job satisfaction is predicted by positive spillover from work to family, characteristics that facilitate work and family—like support from family members and freedom of choice, for example—do so by being linked to fewer negative and more positive spillover effects between remote work and family [98]. On the other hand, there is a less positive and negative correlation between job and family when it comes to work pressure and family conflicts [84].

In the context of remote work, job satisfaction and employee performance are now factors that reduce work–family conflicts [99]. Furthermore, the job happiness of employees who work remotely is impacted by factors such as restricted communication, work–family balance, institutional and technological assistance, and job satisfaction [100]. Additionally, it has been observed that in a remote work setting, institutional and technical support affect work–life balance [101]. Lastly, counterproductive results showed that work–life balance is negatively impacted by remote work but not job satisfaction; similarly, personality traits related to a job do not encourage conflicts of interest between private parties; rather, contracts improve work–life balance and job satisfaction, and remote workers report higher levels of happiness [68]. 

Similarly, they discovered that psychological discomfort related to the workplace acts as a mediator between the positive effects of work addiction on work–family conflict. Additionally, supportive supervisor conduct reduces the indirect impact of overwork on work–family conflict, and this effect is amplified at high levels of support [102]. On the other hand, they observe that a manager’s propensity to assist their family is influenced by their gender as well as contextual factors like family strife, the workload of their subordinates, and the norm of leader–subordinate contact [13].

They also point out that relationships between supervisor family-supportive behaviors and results, such as performance and work intentions, depend on the national climate [18]. Based on the above, it is intended to respond to the following hypothesis:

**H6.** 
*WFPS has a mediating role in the relationship between FSSB and job satisfaction in remote work.*


#### 1.1.8. FSSB, WLB, and Job Satisfaction in Remote Work

During the workday, it is important to feel support from superiors; in this framework, the FSSB is key to achieve certain objectives, so it is evident that FSSBs positively influence WLB and job satisfaction, where this relationship is moderated by previous telework experience and the WFPS [2]. Likewise, when FSSB interacts with WLB and job satisfaction, it moderates the relationship between both variables and, in turn, impacts job performance [47]. Furthermore, WLB is reported to mediate the relationship between FSSB and job performance [57]. Furthermore, the literature suggests that FSSB and WLB have a positive relationship with job satisfaction [48,103,104].

The terms FSSB, WLB, and job satisfaction in the context of telework are important concepts when analyzing the experience of employees who perform their work from remote locations [2]. FSSB refers to supervisor behavior that supports and promotes employees’ work–life balance [47]. Supervisors who demonstrate FSSB offer flexibility in managing schedules, show understanding of family responsibilities, and promote a work environment that allows employees to attend to family obligations without feeling pressured [105].

For its part, WLB refers to the ability of employees to effectively manage their work and personal responsibilities [47]. In the context of telework, WLB implies the ability to separate time and space devoted to work from activities outside work, which contributes to a higher quality of life [1].

From this perspective, job satisfaction refers to the degree to which employees feel satisfied with their work, including aspects such as the nature of the tasks, work relationships, pay, and the work environment [41]. In the context of telework, job satisfaction is an important indicator of workers’ well-being and can influence their performance and commitment [2,41,42,43,44,47,49,52,57]. When it comes to teleworking, work–life balance is crucial, as the boundaries between work and personal life can become blurred. The existence of supervisors who offer support for family responsibilities can have a significant impact on employees’ job satisfaction, as it contributes to effective WLB management [42]. A healthy work–life balance can increase the job satisfaction of telecommuting employees, which in turn can positively impact their performance and well-being [48].

FSSB is very important in ensuring WLB and achieving both worker satisfaction and organizational goals, especially in home-based work conditions [47]. In this way, it has been shown that supervisor behaviors that favor harmony between work and family life reduce work pressure and allow for more autonomous and responsible work on the part of the worker, which ultimately translates into well-being at work and worker commitment to the organization [105]. The flexibility and independence generated by the FSSB help to reduce work–family conflict, especially when the employee is in remote working conditions, since increasing the employees’ control over their work and allowing them to find a balance between their work and family life results in high employee identification with the company that provides these benefits [106]. Likewise, employees who believe that their managers care about their personal and professional lives are more likely to improve their performance and meet supervisory objectives [2].

Work–family reconciliation is of great importance to ensure a balance between remote work and family life, and to achieve worker satisfaction and company goals when the worker is not under the direct supervision of face-to-face work. Research has shown that supervisor actions that promote work–family harmony have the effect of decreasing work-related stress and facilitating greater motivation for work activities performed in the office or remotely [17,48,105,107]. Consequently, this leads to higher job satisfaction and greater employee dedication to the organization. Similarly, work–family conflict can be mitigated by the greater flexibility and autonomy provided by FSSB. This practice empowers employees by giving them greater control over their work responsibilities, allowing them to effectively manage their professional obligations while attending to their family commitments [47,48]. Thus, remote workers can achieve harmony between their work and family life, since there is a positive correlation between employees’ perception of supervisors’ concern for their personal and professional lives and the likelihood of improving their performance, increasing satisfaction, and achieving business goals. 

This affirms that, during the workday, it is important to feel the support of superiors; in this framework, FSSB is key to achieving certain objectives. Therefore, it is evident that FSSB positively influences WLB and job satisfaction, where this relationship is moderated by previous telework experience and WFPS [2]. Likewise, when WFPS interacts with WLB and job satisfaction, it moderates the relationship between both variables and, at the same time, influences job performance [47]. Furthermore, it is specified that WLB mediates the relationship between WFPS and job performance [57]. Furthermore, the literature suggests that FSSB and WLB have a positive relationship with job satisfaction [48,103,104]. Based on the above, we attempted to answer the following hypothesis:

**H7.** 
*WLB has a mediating role in the relationship between FSSB and job satisfaction in remote work.*


The hypotheses of this study are graphically represented in Figure 1 in accordance with the previous paragraphs.

## 2. Materials and Methods

The objective of this research is to propose and validate a theoretical model that ex-plains job satisfaction in remote work, influenced by family-supportive supervisory behaviors (FSSBs) and, in addition, to evaluate the mediating role of work-to-family positive spillover (WFPS) and work–life balance (WLB) in this influence. This research was conducted using a quantitative methodology and employed a non-experimental, cross-sectional design. Data collection involved the administration of a self-administered questionnaire [108].

### 2.1. Sample and Procedure

A non-probability convenience sampling method was used to collect the data for this study [109]. In order to conduct the study, an online survey was administered via the Google form platform, with the survey link disseminated through the WhatsApp messaging service. The poll was conducted in the city of Lima, the capital of Peru, between 13 May 13 and 21 June 2022. The study was centered on those who reported engaging in remote work.

In order to encourage survey participation, the respondents were provided with information regarding the voluntary nature of their involvement, the anonymous analysis of acquired data, and the sole academic utilization of such data. By employing this method, a total of 396 questionnaires were successfully retrieved. The age group with the highest number of participants consisted of individuals between the ages of 31 and 45. This group was predominantly unmarried, had attained a university education, had been employed for a period of 1 to 4 years, and did not have any children residing in their household (see Table 1).

### 2.2. Measurements

In order to construct the research model, the assessment instrument developed by Jamal et al. (2021) was utilized [2]. This instrument was specifically designed to assess job satisfaction in the context of the COVID-19 epidemic. The instrument comprises a total of 16 items, which are allocated to assess four variables: family-supportive supervisory behaviors (consisting of 3 items), work-to-family positive spillover (consisting of 4 items), work–life balance (consisting of 4 items), and job satisfaction in remote work (consisting of 5 items) [2]. The evaluation of all items is conducted using a Likert-type scale, which encompasses a range of 1 to 5 points. In this scale, a rating of 1 indicates complete disagreement, while a rating of 5 signifies complete agreement.

The digital questionnaire was partitioned into two distinct sections. The initial portion of the study encompassed a comprehensive compilation of 16 elements pertaining to the suggested theoretical framework. The subsequent component consisted of a series of inquiries designed to gather sociodemographic information, including age, gender, marital status, educational attainment, duration of employment within the organization, and the number of dependent children residing in the household.

### 2.3. Statistical Analysis

The hypotheses were tested using partial least squares structural equation modeling (PLS-SEM) in the context of data analysis. Partial least squares structural equation modeling (PLS-SEM) is a robust statistical analysis method that incorporates both measurement and structural components. It allows for the simultaneous examination of relationships between variables within a conceptual model. Notably, PLS-SEM is particularly suited for multivariate analysis, where the number of variables involved is equal to or greater than three [109]. Moreover, the present study employed the partial least squares structural equation modeling (PLS-SEM) technique because of its ability to assist the creation of theoretical frameworks [110]. The PLS-SEM analysis was conducted using SmartPLS (Version 4.0.9.5). The utilization of this software was motivated by multiple factors. First and foremost, it is widely regarded as an optimal selection when researchers endeavor to validate a pre-existing theory [111]. Furthermore, it is worth noting that exploratory research often involves intricate models that are best analyzed using partial least squares structural equation modeling (PLS-SEM) [112]. In contrast to a segmented approach, PLS-SEM adopts a holistic perspective by examining the model as a whole [113]. Additionally, PLS-SEM offers simultaneous analysis for both the structural model and the measurement, leading to precise and reliable estimations [114].

## 3. Results

The evaluation of a model using partial least squares structural equation modeling (PLS-SEM) is a two-step procedure that encompasses the assessment of both the measurement and structural models [110,115]. The evaluation of the measuring model encompasses an examination of the constructs’ validity and reliability. This stage entails an assessment of the correlation between each construct and its corresponding components, specifically the replies to the individual question statements in the questionnaire. The evaluation of structural models pertains to the examination of the interrelationships among constructs [110,115]. 

### 3.1. Analysis of Reliability and Validity

In order to evaluate the efficacy of reflective constructs, it is imperative to examine the convergent validity and reliability of the notion, particularly with regard to its internal consistency. In order to ascertain the appropriateness of convergent validity, it is imperative that the loading of each indicator surpasses the threshold of 0.7 [110]. The reliability of the construct is evaluated by Cronbach’s alpha coefficient (α) and the composite reliability coefficient (CR), which must be equal to or greater than 0.70 [115,116,117]. Additionally, the average variance extracted (AVE) must be analyzed, which must be above the threshold of 0.5 [110,115].

In this study’s four reflective constructs, each element’s loading is shown in Table 2 to be larger than 0.7. Additionally, Cronbach’s alpha and the CR both displayed values greater than 0.8, showing the strong reliability of the measurement model. Additionally, the AVE of the constructs was higher than 0.6, demonstrating the measuring model’s outstanding convergent validity.

The Cronbach’s Alpha (α) values for all variables exceed 0.8, indicating a high level of internal consistency. Additionally, the composite reliability (CR) values are above 0.70, suggesting good reliability. The mean variance extracted (AVE) values are greater than 0.50, indicating that a substantial amount of variance is captured by the constructs. Furthermore, the variance inflation factor (VIF) values are below five, indicating that multicollinearity is not a concern. The *p*-value, which is less than 0.001, demonstrates statistical significance at the chosen level of significance. These results collectively support the validity of the model.

Discriminant validity is a measure that assesses the degree to which each construct within a model is unique from other constructs, and the amount to which there is little overlap in the meaning of indicators that do not belong to multiple constructs [115,118]. According to the literature, it is recommended that the square root of the average variance extracted (AVE) for each construct should exceed the strongest correlation observed between that construct and other constructs in the AVE model [110,115]. The root square of the average variance extracted (AVE) for the constructs and the correlations between the constructs are presented in Table 3. These findings suggest that the model possesses satisfactory discriminant validity. The SmartPLS 4.0.9.5 software is capable of calculating full collinearity for all constructs, enabling the simultaneous evaluation of both vertical and lateral collinearity among constructs [117]. According to Table 2, the level of complete collinearity for the constructs was found to be below five. This finding aligns with the acceptable collinearity criterion for PLS-SEM suggested by Hair et al. and Kok, based on factors [117,119].

### 3.2. Analysis of the Structural Model

In order to assess the structural model, it is necessary to examine and document two initial criteria: the statistical significance of the path coefficients and the coefficient value of R2 for endogenous constructs. Every hypothesis within the context of the structural model is connected to a causal link, which serves to illustrate the connections between two constructs. The path coefficients for each relationship in the model have been computed, along with their corresponding *p*-values. While it is essential for the path coefficients to have statistical significance, the magnitude of the R2 coefficient is heavily contingent upon the specific domain of study. Chin proposes that the values of 0.67, 0.33, and 0.19 might be considered as significant, moderate, and weak indicators of the variable R, respectively [120]. In the field of behavioral investigations, it is widely acknowledged that an R2 value of 0.2 is generally seen as adequate, as indicated in previous research [116,119]. In the current investigation, the R2 coefficients for the variables WFPS, WLB, and JS were determined to be 0.195, 0.369, and 0.727, respectively. Hence, all R2 values exhibited reasonably satisfactory and elevated levels. The findings of this study indicate that the study variables explain a substantial proportion of the variability in JS.

The structural model fit indication, specifically the SRMR value of 0.081, closely aligns with the recommended threshold of 0.080. Consequently, this confirms that the model’s fit is satisfactory [121].

The results show that FSSB has a direct effect on both JS and WFPS and WLB. Therefore, hypotheses H1, H2, and H3 are accepted. Furthermore, it shows that WFPS and WLB have positive effects on JS, allowing hypotheses H4 and H5 to be accepted. Likewise, the results show that WFPS and WLB have a mediating role in the influence of FSSB on JS, accepting hypotheses H6 and H7 (see Figure 2 and Table 4).

Regarding the results found, the analysis carried out using PLS-SEM through the SmartPLS software (Version 4.0.9.5) indicated that FSSB had a positive and significant influence on job satisfaction in remote work (β = 0.359, *p* = 0.000), so hypothesis H1 was accepted. This finding highlights that supervisory actions that are aimed at facilitating family responsibilities are associated with greater employee satisfaction with remote work. Likewise, it was found that FSSB had a positive and significant influence on the WFPS (β = 0.359, *p* = 0.000) and WLB (β = 0.580, *p* = 0.000), so hypotheses H2 and H3 are accepted. This discovery allows us to recognize that a company’s consideration of its employees’ families enhances the indirect impacts of organizational decisions in the family context. Additionally, it fosters a more favorable balance between work and family obligations, which translates into greater job satisfaction and a greater sense of dedication to the organization. Likewise, both the WFPS (β = 0.195, *p* = 0.000) and the WLB (β = 0.519, *p* = 0.00) affect job satisfaction in remote work. This allowed for the acceptance of hypotheses H4 and H5, confirming the relevant role of achieving adequate work–family interactions in the satisfaction levels of remote workers. Finally, the WFPS (β = 0.071, *p* = 0.002) and WLB (β = 0.301, *p* = 0.000) demonstrated a mediating role in the influence of FSSB on job satisfaction in remote work. This finding allows us to recognize that the positive indirect effects of work on the family environment, as well as the optimal balance between family and work commitments, contribute to the improvement of family-supportive behaviors, which ultimately leads to greater job satisfaction among employees. In particular, work–life balance is the main mediator in this causal association.

## 4. Discussion

The sudden transition to telecommuting during the COVID-19 pandemic posed a number of challenges for both workers and organizations [1,20]. This sudden shift had a significant impact on critical variables such as job satisfaction, work–life balance, and employee well-being [21,22,23]. In this context, understanding the organizational factors that can mitigate the negative effects of mandatory telecommuting and improve employee satisfaction has become a topic of growing interest in both academia and practice.

The present study focused on shedding light on these crucial aspects. In particular, it sought to examine the impact of family-supportive supervisory behaviors on the levels of job satisfaction reported by telecommuting workers. In addition, the mediating role of positive work–family spillover and work–life balance on this influence was investigated. That is, we sought to provide a solid empirical basis for a deeper understanding of the mechanisms underlying remote job satisfaction, taking into account the dynamic interaction between the spheres of work and family [47,48,52].

Meanwhile, this study makes a valuable contribution to the literature by shedding light on how family-supportive supervisory behaviors may influence job satisfaction in the context of telecommuting, highlighting the mediating role of crucial factors [54,55,56]. The practical implications derived from these findings can guide organizations in promoting a more satisfying and equitable work environment for their telecommuting employees, which, in turn, can have a positive impact on their performance and well-being.

Based on the results, we can confidently state that supervisor family support exerts a positive influence on job satisfaction in the context of telecommuting. In line with this premise, the results of this study convincingly support this causal relationship, demonstrating a significant and positive effect of supervisor family support on job satisfaction. This finding aligns consistently with the existing literature, which has reported that employees who perceive stronger support from their supervisors in managing their family responsibilities also experience higher job role satisfaction [47,52,59,60,61]. A longitudinal study, for example, has corroborated that family support provided by a supervisor predicts a sustained increase in job satisfaction over time [50].

From a theoretical point of view, the job enrichment model [42] suggests that family support provided by supervisors enhances satisfaction by making it easier for employees to balance work and family demands. In addition, the social exchange theory [55,56] points out that supportive actions generate a feeling of reciprocity in employees, which strengthens their commitment and job satisfaction. Therefore, the family-supportive behaviors of supervisors will always lead to positive effects on workers’ well-being.

Furthermore, the results of this study corroborated the significant and positive effect of the FSSB variable on WFPS, the result of which is consistent with previous research suggesting that supervisors’ actions to facilitate employees’ family responsibilities favor the transfer of positive aspects from work to home [59]. From a theoretical basis, the theory of work–family boundaries [28] suggests that family support from the supervisor helps employees better manage the border between both domains, allowing for the overflow of positive emotions and experiences, thus demonstrating the usefulness and validity of this theoretical model to explain relationships in similar variables. Likewise, one study found that supervisor family support predicted higher levels of positive spillover from work to family in employees with higher organizational commitment. That is, the authors explain that this effect is achieved when a sense of identification is presented to the company [66]. Therefore, the results obtained provide evidence that is consistent with previous studies on the favorable effect of family-supportive supervisory behavior to facilitate positive spillover from work to the home environment among remote workers, thus improving their general well-being.

Likewise, FSSB has a positive influence on WLB in the context of remote work. The results of this study corroborated this relationship, finding a significant and positive effect of the supervisor’s family support variable on work–life balance. These findings are consistent with previous studies that have reported that supervisors’ actions aimed at helping their employees manage their family responsibilities are associated with a better ability to achieve work–life balance [50]. For example, some authors found in a sample of New Zealand employees that those who perceived greater family support from their supervisors had lower levels of work–family conflict and greater work–family facilitation, with both related to a better balance [122]. Consulting theory, the job demands–resources model proposes that resources from supervision, such as family support, help employees cope with the demands of work and family, thus contributing to a better balance between both domains [123]. Likewise, the conservation of resources theory suggests that supervisors’ actions to facilitate family responsibilities represent valuable resources that employees seek to obtain and conserve, thereby reducing conflict and improving work–family balance [124]. That is, the results obtained provide further empirical evidence on the positive effect of family-supportive supervisory behavior on employees’ ability to achieve a healthier work–life balance in the context of remote work.

Another finding refers to the fact that the WFPS positively influences job satisfaction in remote work. This result corresponds to previous studies which report that a negative work–family overflow can affect life satisfaction and is relevant to prevent burnout [78,79]. Likewise, leaders who prioritize and promote balance between their employees’ work and personal life have the ability to decrease their employees’ intention to quit and improve their level of job satisfaction.

Likewise, this study found that WLB positively influences job satisfaction in remote work. This evidence is corroborated by the antecedents that have related job satisfaction with WLB, family–work conflict, and perceived performance [86,87,89]. Furthermore, the reconciliation of work and family life is related to job satisfaction and commitment, with the exception of work stress. This is due to the fact that the impact of work–life balance on stress and job satisfaction is influenced by the work environment [90]. The impact of relationships on job satisfaction is moderated by work–life balance, while job satisfaction improves if role overload is reduced [91]. From an alternative point of view, although remote work does not directly influence job satisfaction, it could have an adverse effect on work–life balance [68].

Furthermore, the results revealed sufficient evidence on the mediating role of WFPS in the influence of FSSB on job satisfaction in remote work. The results presented in this study coincide with the conclusions drawn in previous research, which has also explored the mediation and influence interactions between these variables [2,102]. A manager’s supportive behavior can be influenced by various aspects, such as his or her level of empathy, situational circumstances such as family conflict, subordinates’ work, and contact between leader and subordinate. In addition, it has been observed that gender and family conflicts at work can also influence the manager’s empathy and supportive behaviors [13].

Finally, WLB was found to have a mediating role in the influence of FSSB on job satisfaction in remote work. This mediating role in the variables described corroborates what was found in previous studies, where it has also been related to other variables such as performance, productivity, and work commitment [2,47,48,57,103].

The findings of this study enhance the current body of knowledge by offering empirical evidence on the causal connection between family-supportive supervisory behaviors (FSSBs) and job satisfaction in remote employment. This relationship is further mediated by work-to-family positive spillover (WFPS) and work–life balance (WLB). Nevertheless, it is crucial to recognize the inherent constraints of this research. It is crucial to acknowledge that the data were gathered throughout the COVID-19 epidemic, a time characterized by exceptional circumstances that may have impacted the views and experiences of teleworkers. The dynamics of work are in a constant state of evolution, emphasizing the persistent requirement to examine and assess the impact of these aspects on job satisfaction within a dynamic telework setting.

An additional constraint pertaining to the study participants is the absence of categorization based on workers’ positions within the organizational hierarchy of the companies. Further investigation is necessary to explore this aspect, as potential disparities may exist between individuals in the roles of employee and manager.

One additional constraint of this study is the absence of differentiation across the business sectors of the participants. This limitation arises from the primary focus of the sample’s inclusion criteria, which primarily targeted those engaged in distant labor situations. Hence, it is imperative that next research endeavors to examine potential disparities in the associations between the variables of investigation based on distinct business sectors.

In conclusion, it is crucial to acknowledge the limitations of employment status when interpreting the findings of this study. Furthermore, it is imperative to recognize that the implications and recommendations drawn from this research should be applied within the specific framework of each business, taking into account its distinct work culture and the working conditions experienced by its employees. The implementation of strategies aimed at enhancing remote worker satisfaction might significantly differ based on the unique circumstances of individual companies, highlighting the importance of addressing these concerns in a flexible and tailored approach.

Consequently, drawing upon the findings and discourse elucidated in this investigation, we posit that the subsequent avenues for future inquiry can be suggested: The results indicate that WLB is the main mediator in the causal association between FSSB and job satisfaction in remote work. Therefore, additional research is necessary to thoroughly investigate the mechanisms that underlie this mediation, taking into account contextual factors such as company culture and employee support programs, among other variables. In order to investigate the varying impact of gender on the correlation between FSSB, WFPS, and job satisfaction, it is important to include past research indicating potential gender-based differences in the advantages derived from supervisor support in achieving work–family balance [85]. In order to establish causal links and obtain stronger evidence on the impact of WFPS over time, it is necessary to conduct a longitudinal analysis on the association between WFPS and job satisfaction in remote work. This approach is preferred over cross-sectional studies, which have limits in establishing causality. In conclusion, the findings of this study present many avenues for future research aimed at enhancing the comprehension of organizational and psychosocial determinants that forecast job satisfaction within the novel framework of distant work prompted by the COVID-19 pandemic. There exists a substantial area that warrants further investigation from both a theoretical and empirical standpoint.

## 5. Conclusions

The objective of this research is to propose and validate a theoretical model that explains job satisfaction in remote work influenced by family-supportive supervisory behaviors (FSSBs) and, in addition, to evaluate the mediating role of work-to-family positive spillover (WFPS) and work–life balance (WLB) in this influence. The findings of this study provide robust evidence that FSSB has a substantial and statistically significant impact on job satisfaction in the context of remote work. In instances where managers demonstrate actions that support and enable employees in fulfilling their family duties, employees tend to have higher levels of satisfaction with their remote work arrangements. Similarly, research has indicated that FSSB has a strong beneficial impact on WFPS and WLB. In other words, when a company prioritizes the well-being of its employees’ families, it enhances the equilibrium between work and family life and facilitates the transfer of positive elements from the work setting to the family setting. In contrast, it is evident that both WFPS and WLB have a favorable impact on the level of satisfaction experienced by individuals engaged in distant employment. This underscores the significance of attaining a satisfactory work–family interface in relation to the levels of satisfaction experienced by individuals engaged in teleworking.

This study provided evidence for the mediating effect of WFPS and WLB on the association between FSSB and remote job satisfaction. This observation highlights the need for acknowledging the indirect positive impacts of employment on the family unit, as well as the ideal equilibrium between familial and occupational obligations. These factors collectively amplify the influence of a supervisor’s supportive actions towards the employee’s family on their overall job satisfaction.

In summary, this research offers robust empirical support to validate the notion that FSSB is a substantial determinant of job satisfaction in the context of remote work. Furthermore, this effect is amplified by the attainment of a satisfactory equilibrium between work and family responsibilities. The findings of this study have significant practical significance for firms seeking to enhance the satisfaction and fairness of remote work settings.

## Figures and Tables

**Figure 1 behavsci-13-00916-f001:**
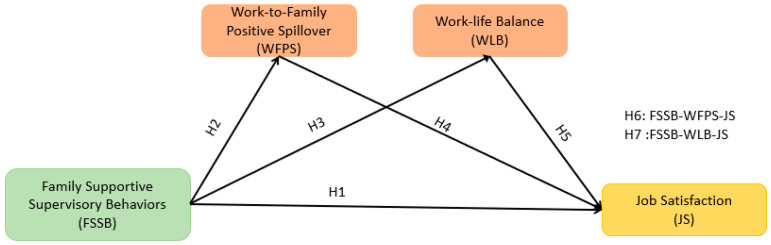
Theoretical research model.

**Figure 2 behavsci-13-00916-f002:**
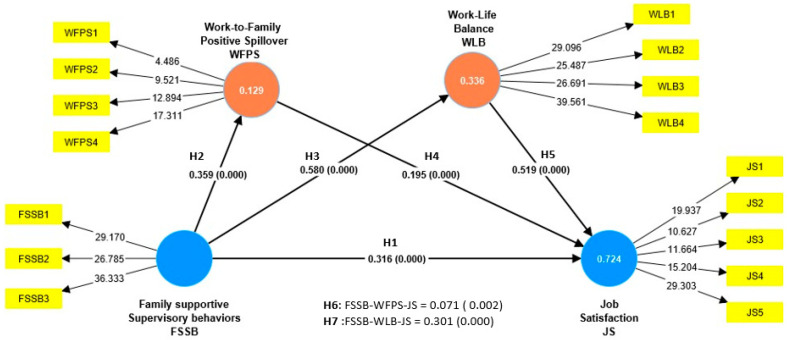
Results of the structural model.

**Table 1 behavsci-13-00916-t001:** Sociodemographic data of the participants (n = 396).

	Female		Male		Total	
Age range	n	%	n	%	n	%
18–30	80	20.20	46	11.60	126	31.80
31–43	128	32.30	67	16.90	195	49.20
44–56	43	10.90	19	4.80	62	15.70
57–70	5	1.30	8	2.00	13	3.30
Total	256	64.60	140	35.40	396	100.00
**Civil Status**			**Academic Level**			
	n	%			n	%
Married	113	28.50	Postgraduate		99	25.00
Cohabitant	49	12.40	Secondary		17	4.30
Divorced or widowed	17	4.30	Advanced technician		95	24.00
Single	217	54.80	academic		185	46.70
Total	396	100.00	Total		396	100.00
**Seniority in the Workplace**			**Number of Children at Home**			
	n	%			n	%
1 to 4 years	140	35.40	1 child		96	24.20
5 to 10 years	97	24.50	2 children		77	19.40
More than 10 years	88	22.20	3 children or more		30	7.60
Less than 1 year	71	17.90	None		193	48.70
Total	396	100.00	Total		396	100.00

**Table 2 behavsci-13-00916-t002:** Validation of the measurement model (reliability and convergent validity).

Construct	Code	Loading	*p*-Value	(α)	C.R.	AVE	VIF
Family-Supportive Supervisory Behaviors	FSSB1	0.927	<0.001	0.918	0.919	0.860	3430
	FSSB2	0.920	<0.001				3004
	FSSB3	0.935	<0.001				3579
Job Satisfaction	JS1	0.788	<0.001	0.856	0.865	0.633	1654
	JS2	0.725	<0.001				1637
	JS3	0.810	<0.001				2352
	JS4	0.840	<0.001				2469
	JS5	0.812	<0.001				1851
Work–Life Balance	WLB1	0.843	<0.001	0.918	0.919	0.804	2057
	WLB2	0.909	<0.001				3552
	WLB3	0.915	<0.001				3773
	WLB4	0.918	<0.001				3937
Work-to-Family Positive Spillover	WFPS1	0.723	<0.001	0.841	0.892	0.666	1992
	WFPS2	0.836	<0.001				2390
	WFPS3	0.863	<0.001				2171
	WFPS4	0.836	<0.001				1645

Source: Self-elaboration. Note: (α) = Cronbach’s Alpha; C.R = composite reliability; AVE = average variance extracted; VIF = variance inflation factor.

**Table 3 behavsci-13-00916-t003:** Discriminant validity.

	FSSB	JS	W.L.B.	WFPS
FSSB	**0.927**			
JS	0.613 ***	**0.796**		
WLB	0.533 ***	0.697 ***	**0.897**	
WFPS	0.325 ***	0.457 ***	0.369 ***	0.816

**Note.** The square root of AVEs is shown diagonally in bold. *** (significance level < 0.001).

**Table 4 behavsci-13-00916-t004:** Results of hypotheses testing.

H	Hypothesis	Pat Coefficient	*p*-Value	Decision
H1	FSSB → JS	0.316	0.000	Accepted
H2	FSSB → WFPS	0.359	0.000	Accepted
H3	FSSB →WLB	0.580	0.000	Accepted
H4	WFPS → JS	0.195	0.000	Accepted
H5	WLB → JS	0.519	0.000	Accepted
H6	FSSB →WFPS → JS	0.071	0.002	Accepted
H7	FSSB → WLB → JS	0.301	0.000	Accepted

## Data Availability

The data may be provided to interested readers by requesting the corresponding author via email.

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
