# Peer review of "Job Satisfaction in Remote Work: The Role of Positive Spillover from Work to Family and Work–Life Balance"

_behavsci, 2023, doi:10.3390/bs13110916_

Round 1

Reviewer 1 Report

Comments and Suggestions for Authors

The manuscript deals with an interesting and topical subject. However, the following observations are made:

1.1. Literature Review and Hypothesis 

Section 1.1.8 should be summarized and more focused on job satisfaction and remote work

Paragraphs 1.1.7 and 1.1.8 should be reinforced by more literature support.

Materials and Methods

It is of interest to introduce the category of employees into the study. For example, for the purposes of the study, there may be differences between an employee and a manager.

It is necessary to indicate to which sector or sector the survey was directed.

Discussion

The Discussion section should be improved.

This section indicates that the hypotheses corroborate what has been previously investigated in scientific studies. It is important to indicate what the manuscript contributes to the literature through the different hypotheses tested.

It is recommended to make clear what added value the manuscript brings to the literature and the more concrete practical implications for organisations.

The Conclusions section may be improved as a result of possible modifications in the Discussion section.

References

I would like to suggest two publications from this year that should be included in the manuscript.

Kim, M.(S.), Ma, E., & Wang, L. (2023). Work-family supportive benefits, programs, and policies and employee well-being: Implications for the hospitality industry.  International Journal of Hospitality Management, 108.

https://doi.org/10.1016/j

Medina-Garrido, J.A., Biedma-Ferrer, J.M., Bogren, M. (2023). Organizational support for work-family life balance as an antecedent to the well-being of tourism employees in Spain,  Journal of Hospitality and Tourism Management, 57, 117-129, https://doi.org/10.1016/j.jhtm.2023.08.018. 

Author Response

Dear Reviewers,

Thank you very much for your informed comments, which helped us so much in improving the manuscript.  We appreciated the time you spent in doing this and tried our best to address all your comments.

We hope that this revised version of the paper reaches the expected standard, worthy of publication in this journal.

A detailed list of answers to your comments and suggestions is reported below.

Many thanks for your time.

Best regards,

Reviewer 2 Report

Comments and Suggestions for Authors

The manuscript with the title: "Job Satisfaction in remote work: The Role of Positive Spillover from Work to Family and Work-Life Balance" is interesting and actual. Researching job satisfaction is very important in the time of digital transformation. Job satisfaction and the work-life balance between private life and work life were significantly affected during the COVID-19 pandemic, so I consider the submitted manuscript to be relevant. Results of the research can be used in the development of individual performance of employees as well as in the development of organizational performance. Overall, I rate the submitted contribution as excellently processed. I have only a few small recommendations for authors that can help improve the submitted manuscript before it is published:

-          I recommend editing the abstract (unify the main aim), indicate the size of the research file and remove or edit the last sentence: However, it is recognized that these results are based on data collected during the COVID-19 pandemic, so there is a need to continue investigating how these factors affect job satisfaction in an ever-evolving remote work environment.

-          It is necessary to unify the main aim of the article (line 281-282): Therefore, this study aims to answer the following hypotheses: H2. The FSSB has a positive influence on the WFPS in remote work. (line 110-113): Based on the above, the objective of this research is to propose a theoretical model that explains job satisfaction in remote work influenced by family supportive supervisory behaviors and, in addition, to evaluate the mediating role of Work-to -family positive spillover and work-life. balance in this influence. (line 15-16): This study focuses on analyzing job satisfaction in the context of remote work, focusing on key factors such as supervisor behavior, positive work-to-family spillover, and work-life balance. The main aim of the article and the main aim of the research should be explicitly defined and uniform throughout the manuscript.

-          1.1. Literature Review and Hypothesis – I appreciate the processing of the mentioned part (line 118-424), but I recommend selecting the processed text, or to expand it with studies and research carried out expressly from the period of the COVID-19 pandemic. Not all the resources used are up-to-date and from the period of the pandemic.

-          Table 1, the interval of years should be the same and not different, for example 15 years and so... Seniority in the workplace - less than 1 year – need to be moved from 1 to 4 years ago.

-          The last recommendations for improvement are about the conclusions, I think the authors should expand the limitations of the research as well as future research. Further, the authors mention recommendations for practice (line 698-701: Ultimately, the implications and recommendations derived from this study must be considered in the context of the uniqueness of each organization and its work culture, since strategies to improve remote worker satisfaction may vary depending on the specific circumstances of each company.), but we do not encounter them in the article, so it would be appropriate to supplement them in a brief form.

Overall, I rate the article well prepared and qualitatively methodically processed! I wish the authors good luck in their research.

Author Response

(The authors gave the same response as above.)

Reviewer 3 Report

Comments and Suggestions for Authors

Revise wording of 59-61.

Revise wording line 64.

Statements in paragraph 72-78 require references.

Revise wording line 79.

Delete 113-117.

Revise wording of 435-437.

Revise data format of Table 1 (tabulation and capitalization).

Delete period in Jamal et al line 456.

Line 506 refers to "nine reflective constructs" but Table 2 lists 4.

Table 2 needs explanatory table footnote.

The SRMR value is given but no further details are given for the other absolute and relative fit indices.

Paragraph 587-607 should be presented in results. 

Comments on the Quality of English Language

Fine

Author Response

(The authors gave the same response as above.)
